# CaSiO_3_-HAp Metal-Reinforced Biocomposite Ceramics for Bone Tissue Engineering

**DOI:** 10.3390/jfb14050259

**Published:** 2023-05-08

**Authors:** Evgeniy K. Papynov, Oleg O. Shichalin, Anton A. Belov, Igor Yu Buravlev, Vitaly Yu Mayorov, Alexander N. Fedorets, Anastasiya A. Buravleva, Alexey O. Lembikov, Danila V. Gritsuk, Olesya V. Kapustina, Zlata E. Kornakova

**Affiliations:** Far Eastern Federal University, 10 Ajax Bay, Russky Island, 690922 Vladivostok, Russia

**Keywords:** selective laser melting, spark plasma sintering, wollastonite/hydroxyapatite, bone tissue, scaffolds

## Abstract

Reconstructive and regenerative bone surgery is based on the use of high-tech biocompatible implants needed to restore the functions of the musculoskeletal system of patients. Ti6Al4V is one of the most widely used titanium alloys for a variety of applications where low density and excellent corrosion resistance are required, including biomechanical applications (implants and prostheses). Calcium silicate or wollastonite (CaSiO_3_) and calcium hydroxyapatite (HAp) is a bioceramic material used in biomedicine due to its bioactive properties, which can potentially be used for bone repair. In this regard, the research investigates the possibility of using spark plasma sintering technology to obtain new CaSiO_3_-HAp biocomposite ceramics reinforced with a Ti6Al4V titanium alloy matrix obtained by additive manufacturing. The phase and elemental compositions, structure, and morphology of the initial CaSiO_3_-HAp powder and its ceramic metal biocomposite were studied by X-ray fluorescence, scanning electron microscopy, energy-dispersive X-ray spectroscopy, and Brunauer–Emmett–Teller analysis methods. The spark plasma sintering technology was shown to be efficient for the consolidation of CaSiO_3_-HAp powder in volume with a Ti6Al4V reinforcing matrix to obtain a ceramic metal biocomposite of an integral form. Vickers microhardness values were determined for the alloy and bioceramics (~500 and 560 HV, respectively), as well as for their interface area (~640 HV). An assessment of the critical stress intensity factor *K*_I*c*_ (crack resistance) was performed. The research result is new and represents a prospect for the creation of high-tech implant products for regenerative bone surgery.

## 1. Introduction

Synthetic wollastonite (CaSiO_3_) has the highest potential for bone engineering among bone ceramic implants [1]. This is due to its unique set of properties: nontoxicity, bioactivity, antimicrobial effect, structural porosity, low moisture absorption, corrosion resistance, bioresorption, and more [2,3,4,5]. An extensive list of known studies on the biocompatible properties of wollastonites in the world is presented in the research by Biswas et al. [2], and is also reflected in the experimental studies [6]. A review of present studies on the toxicity of wollastonite is presented in [7]. At the same time, along with wollastonite, the prospect of using its biocomposite forms, in particular, the ones reinforced with hydroxyapatite (HAp) has been noted [8,9]. The presence of synthetic HAp in the composition of the biocomposite endows the product with a more complete set of bioactive properties; in particular, its osseointegration efficiency increases. This is proved by relevant studies, including earlier ones where bioactive properties were demonstrated on in vivo models under conditions of healing of jawbone defects [10,11]. Synthetic wollastonite is applied in medicine in the form of implants for reconstructive and regenerative bone surgery in dentistry, orthopedics, maxillofacial plastic surgery, and more [12,13,14].

However, there is an obvious technological difficulty in obtaining volumetric ceramic products of the required geometry, developed structural porosity of the hierarchical type, and high mechanical strength based on wollastonite and its composite reinforced with HAp. Most traditional sintering methods do not allow for achieving these characteristics, especially in their optimal ratio in the product. Rigid technological regimes (high temperature and pressure) lead to the destruction of the composition, in particular, due to the thermal instability of HAp, destruction (sintering) of the porous framework, grain growth, and the occurrence of internal stresses, resulting in, as a consequence, a decrease in the mechanical strength of the biomaterial and its composites [12,15,16,17]. In order to solve the problems in the synthesis of CaSiO_3_ biocomposite ceramics, including those reinforced with HAp, the approaches using an unconventional technique of spark plasma sintering (SPS, widely studied in [18,19,20]), and reactive SPS, as well as combinations of this technique with sol–gel and template syntheses, have been considered and implemented. As a result, micro- and nanostructured CaSiO_3_ bioceramics with a macroporous [21,22] and bimodal ordered structure [23], with doped nanosized gold [24], reinforced with HAp [25,26], which have biocompatible properties, relatively high mechanical characteristics, and a defect-free porous framework have been successfully obtained and studied.

Along with this, in recent studies, scientists have demonstrated the possibility of using the SPS technique to obtain ceramic composites reinforced with cellular titanium alloy matrices (Ti6Al4V, known for medical applications [27,28,29,30]) manufactured by additive 3D printing technologies. Only three studies, all of which were performed by Rahmani et al., are known in this area [31,32,33], where the method of combining the selective laser melting technology and SPS was applied to obtain the following medical biocomposites: Ti6Al4V-TiO_2_/Ag and Ti6Al4V-CaSiO_3_. Antibacterial properties of these biocomposites, as well as their high mechanical characteristics, have been demonstrated and substantiated in detail in these studies. Based on the results of the studies, it has been concluded that this technological approach is promising for creating ceramic–metal composites used for manufacturing implants able to withstand loads, provide high bond strength at the ceramic-metal interface, exhibit biocompatibility, and not be rejected by a living organism.

According to the above, it should be noted that the possibility of using wollastonite (CaSiO_3_) reinforced with synthetic hydroxyapatite (HAp, Ca_10_(PO_4_)_6_(OH)_2_) as a bioceramic base in the considered ceramic–metal composite is a highly promising solution. In addition to the bioactive properties of wollastonite, Hap (a synthetic analog of bone tissue [34,35]), which is used for its reinforcement, also provides high bioactivity in the process of osteosynthesis, which will allow for activating the growth of bone tissue into the cellular Ti6Al4V metal frame. This, in turn, may enhance the osseointegration properties of the ceramic metal implant. The aim of this research was to study the possibility of obtaining CaSiO_3_-HAp biocomposite ceramics using SPS technology and reinforcing them with a Ti6Al4V titanium alloy matrix obtained by additive manufacturing. The result of the research is new and may represent a high-tech solution for manufacturing appropriate implants for bone regenerative surgery.

## 2. Experimental Part

### 2.1. Reagents

The used reagents were as follows: calcium chloride (CaCl_2_), sodium silicate (Na_2_SiO_3_), ammonium hydrogen phosphate ((NH_4_)_2_HPO_4_), ammonia ((NH_4_OH). The Ti6Al4V titanium alloy reinforcing matrix (ratio 90:6:4 wt.%) was manufactured by selective laser melting technology on a Concept Laser M2 setup (Lichtenfels, Germany) using Rematitan^®^CL metal alloy powder (matrix dimensions: diameter 15 mm, height 15 mm). All chemicals were purchased from Sigma-Aldrich (St. Louis, MO, USA) at 99.9% purity without additional purification.

### 2.2. Method for Synthesizing CaSiO_3_-HAp Biocomposite Powder

Eighty-eight milliliters of 1 M CaCl_2_ solution, 80 mL of 1 M Na_2_SiO_3_, 5 mL of 1 M (NH_4_)_2_HPO_4_, 7 mL of 25% NH_4_OH solution, and 20 mL of distilled H_2_O were poured into a Teflon beaker. Next, the Teflon beaker was placed in a steel autoclave and heated in an oven at 180 °C for 6 h. After that, the autoclave was cooled, and the resulting precipitate was filtered through a blue tape filter and dried at 100 °C in the air until residual moisture was removed. The resulting material was sieved, and a fraction of the powder with particles 100–200 µm in size was selected. The powder was calcined in a muffle furnace in the air at 800 °C with a heating rate of 10 °C/min and a holding time of 60 min.

The reaction equation was as follows:CaCl_2_ + Na_2_SiO_3_ → CaSiO_3_ + 2NaCl.
10CaCl_2_ + 6(NH_4_)_2_HPO_4_ + 8NH_4_OH → Ca_10_(PO_4_)_6_(OH)_2_ + 20NH_4_Cl + 6H_2_O.

### 2.3. Method for Obtaining CaSiO_3_-HAp Biocomposite Reinforced with Ti6Al4V Matrix

The CaSiO_3_-HAp biocomposite reinforced with a titanium alloy matrix was obtained by SPS technology on an SPS-515S unit from Dr. Sinter·LAB^TM^ (Tsurugashima, Japan) according to the following scheme (Figure 1): 2 g of CaSiO_3_-HAp powder was placed in Ti6Al4V matrices using a CISA RP-200-N (Barcelona, Spain) sieve shaker. Next, the mold was placed in a graphite molding die (working diameter 15.5 mm), pre-pressed (pressure 20.7 MPa), and then installed in a vacuum chamber (10^−5^ atm), after which it was heated. The molding die with the mold was heated by a low-voltage pulsed unipolar current in the On/Off mode, with a frequency of 12 pulses/2 pauses, i.e., the pulse burst duration was 39.6 ms and the pause was 6.6 ms. The sintering temperature was 900 °C, the heating rate was ~300 °C/min in the range up to ~650 °C, then 100 °C/min in the working area of the pyrometer. The specimen was held at the maximum temperature for 5 min and then cooled to room temperature for over 30 min. The pressing pressure throughout the entire process was 21.5 MPa. The molding die was wrapped in a heat-insulating fabric to reduce heat loss during heating. The geometric dimensions of the obtained product specimen were as follows: diameter 15 mm, height 5 mm.

### 2.4. Characterization Methods

Phase identification was conducted by the means of X-ray diffraction (XRD) on a diffractometer D8 Advance “Bruker AXS” (Bremen, Bruker), using CuKα-source, Ni-filter, angle range 10–80°, scanning step 0.02°, scanning rate 5°/min. The specific surface area was measured by low-temperature nitrogen physisorption at 77 K on an automated gas sorption analyzer, Autosorb IQ “Quantachrome” (FL, USA), results were analyzed at the level of Brunauer–Emmett–Teller method (BET), BJH, and DFT models). Surface imaging of the fabricated samples was performed by scanning electron microscopy (SEM) on a CrossBeam 1540 XB “Carl Zeiss” microscope (Oberkochen, Germany) with an energy dispersive X-ray analysis (EDX) add-on by Bruker (Bremen, Germany). In order to compensate for surface charging, a gold layer of up to 10 nm was sputtered on the samples.

Vickers microhardness (HV) was estimated at a force of 0.2 N on an HMV-G-FA-D “Shimadzu” microhardness tester (Kyoto, Japan).

The ISO 13586 standard was used to evaluate the critical stress intensity factor *K*_I*c*_ (crack resistance). These measurements allowed for estimating the specific amount of local stress at the crack tip that caused uncontrolled crack growth and, hence, specimen failure. The value was calculated by the following formula:KIc=f(aw)·Fqh·w
where *f(a/w)* was the geometric factor, *h* and *w* were the specimen thickness and width, respectively, and *F_q_* was the crack initiation load. The index “I” denoted the loading mode I, in which the applied load was perpendicular to the crack plane. The loading mode I was considered the most severe and the most important one.

## 3. Results and Discussion

The CaSiO_3_-HAp biocomposite powder was obtained by precipitation in solution under conditions of hydrothermal heating of the solution, followed by thermal oxidative calcination of the formed precipitate. According to XRD (Figure 2), the composition of the resulting dispersed material included the crystalline phase of wollastonite (CaSiO_3_) combined with hydroxyapatite (Ca_10_(PO_4_)_6_(OH)_2_). Foreign impurities were absent.

According to the SEM (Figure 3), the morphology of the resulting CaSiO_3_-HAp biocomposite powder was characterized by an inhomogeneous structure due to the composite composition. The material consisted of mixed acicular (wollastonite) and monolithic (hydroxyapatite) agglomerates. The structure of the agglomerates was loose, which indicated the presence of porosity in the biocomposite [36]. It is known that silicate disperse systems such as those under study are non-toxic, chemically inert, thermally stable, porous, and have a variety of applications in biomedicine as bioinert/bioactive materials. The synthesis of dense ceramic compacts based on such powders opens up new possibilities for their practical application as bioceramic matrices (implants) for bone surgery [37].

According to the analysis of the porous structure of the CaSiO_3_-HAp biocomposite powder, it was found that the low-temperature sorption–desorption nitrogen isotherm at 77 K corresponded to type IV IUPAC classification curves (Figure 4a). This indicated that the specimen was characterized by the presence of a porous structure with a wide range of micro-, meso-, and macropore size distributions. According to the calculation of the experimental data on the pore size distribution by the Barrett–Joyner–Halenda method and according to the calculations using the density functional theory of electronic structure, it was determined that the specimen did not have a narrow pore size distribution; the pore range ranged from 4 nm to 60 nm or more (Figure 4b,c). The specific surface area of the sample was 4 m^2^/g.

For materials in which the mesopore system consists of large cellular pores interconnected by smaller mesopores and/or embedded in a microporous matrix, such as carbons prepared from 3D colloidal templates (3DOm), the use of a cylindrical–spherical adsorption core is recommended [38]. The use of a cylindrical or slotted model for micro- and mesopores smaller than 5 nm is reasonable, as such pores (if present) are not affected by templating.

According to XRD data, the composition of the initial sintered biocomposite powder did not change under SPS conditions in the region of the studied temperature and was represented by crystalline phases of wollastonite and HAp (Figure 2, curve 2). The SEM images showed that the reinforcing matrix with a uniform cellular structure made of a dense Ti6Al4V alloy [39] (Figure 5a), after being filled with the initial biocomposite powder and heated by SPS, was tightly integrated into the volume of the formed CaSiO_3_-HAp bioceramics (Figure 5b). The shape and size of the cell structure on the surface of the specimen remained practically unchanged, despite the fact that its entire volume underwent a certain deformation and shrinkage (Figure 5b). It can be seen that there were no defects at the bioceramics and alloy interface (Figure 5c). The high efficiency of sintering the initial powder into a dense compound and its homogeneous integration with the reinforcing alloy was evident. The ceramic structure was characterized by the presence of nonpenetrating macropores (Figure 5d). According to the EDX analysis data (Figure 5e), it was established that there was no interdiffusion of the main elements between the ceramics and the alloy. The bioceramics and alloy interface was relatively uniform, and the melting zone was not observed.

Vickers microhardness results reached relatively high values for alloy (~500 HV) and ceramics (~560 HV). According to the scatter diagram, the values of microhardness for the alloy and ceramics had insignificant differences, which indicated the uniformity of the strength of the materials over their volume (Figure 6). A wide scatter of values was observed at the alloy and ceramics interface, which was probably due to internal stresses in this area. Uncompensated mechanical stresses were induced by the difference in the coefficients of linear thermal expansion of dissimilar materials that occurred during heating, which affected their overall mechanical strength. The total value of microhardness at the alloy and ceramics interface was higher (~640 HV). This was probably induced by higher heating of the alloy and, accordingly, of ceramics contacting with the alloy during the passage of an electric current, which affected the local mechanical characteristics of each of the materials.

According to SEM images (Figure 7), distinct indenter marks were observed, where it could be seen that at a selected load of 0.2 H, cracks formed in the volume of ceramics (Figure 7b), including at the alloy and ceramics interface (Figure 7c). The calculated values of ceramic crack resistance were *K*_I*c*_ = 0.304 MPa∙m^1/2^ and *K*_I*c*_ = 0.417 MPa∙m^1/2^ for the interface area, which indicated its brittleness.

Thus, this study demonstrates for the first time that volume-printed porous CaSiO_3_-HAp powder with a Ti6Al4V reinforcing matrix has superiority in both bone regeneration potential and mechanical development in the repair of thin-walled bone defects. In this regard, mechanically robust volume CaSiO_3_-HAp powder with a Ti6Al4V reinforcing matrix is promising for certain scenarios of bone defect repair, especially for the repair of thin-walled craniomandibular-facial bone defects [40,41].

## 4. Conclusions

This research investigates the possibility of using SPS technology to obtain new CaSiO_3_-HAp biocomposite ceramics reinforced with a Ti6Al4V alloy matrix obtained by additive manufacturing. The XRD and SEM methods revealed that the consolidation of the initial CaSiO_3_-HAp powder under SPS conditions at 900 °C and at a holding time of 5 min was accompanied by the formation of dense ceramics, without changing the phase composition, in the volume of which the reinforcing frame of the Ti6Al4V alloy was tightly integrated. The structure of the bioceramics was characterized by the presence of macropores that did not penetrate deep into its volume. The defects in the volume of bioceramics, as well as at the bioceramics and alloy interface, were not observed. According to the SEM and EDX data, melting and diffusion of the main components of the alloy were not defined. It was determined that the alloy and bioceramics had a relatively high Vickers microhardness (~500 and 560 HV, respectively), with a minimum spread of these values over the volume of materials. At the same time, at the bioceramics and alloy interface, the microhardness value was higher (~640 HV), which was induced by the specifics of the heating of the composite by electric current. It was established that the bioceramics had low values of crack resistance in the volume and at the interface area, which were *K*_I*c*_ = 0.304 MPa∙m^1/2^, and *K*_I*c*_ = 0.417 MPa∙m^1/2^, respectively. The research result is new and represents a prospect for the creation of high-tech implant products for regenerative bone surgery.

## Figures and Tables

**Figure 1 jfb-14-00259-f001:**
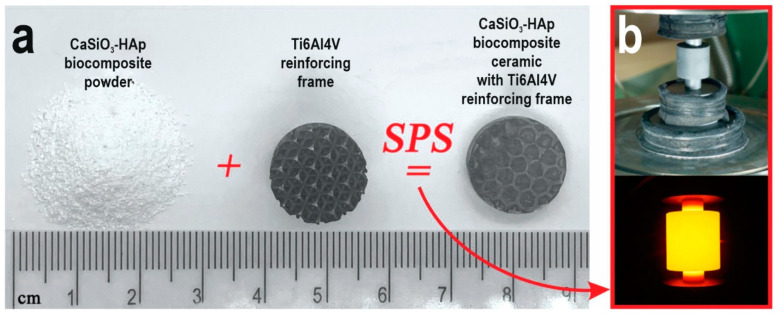
The general scheme of the obtained CaSiO_3_-HAp biocomposite sample, reinforced by Ti6Al4V alloy matrix, by SPS technology: (**a**) the initial blanks and obtained sample; (**b**) SPS process: the mold with the sample assembly and its heating.

**Figure 2 jfb-14-00259-f002:**
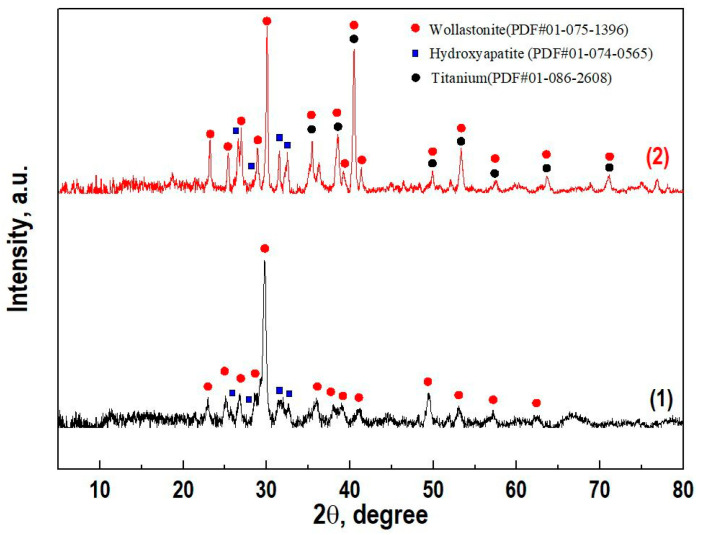
X-ray diffraction patterns of (1) the initial powder CaSiO_3_-HAp biocomposite and (2) ceramics based on it with a reinforced matrix Ti6Al4V alloy.

**Figure 3 jfb-14-00259-f003:**
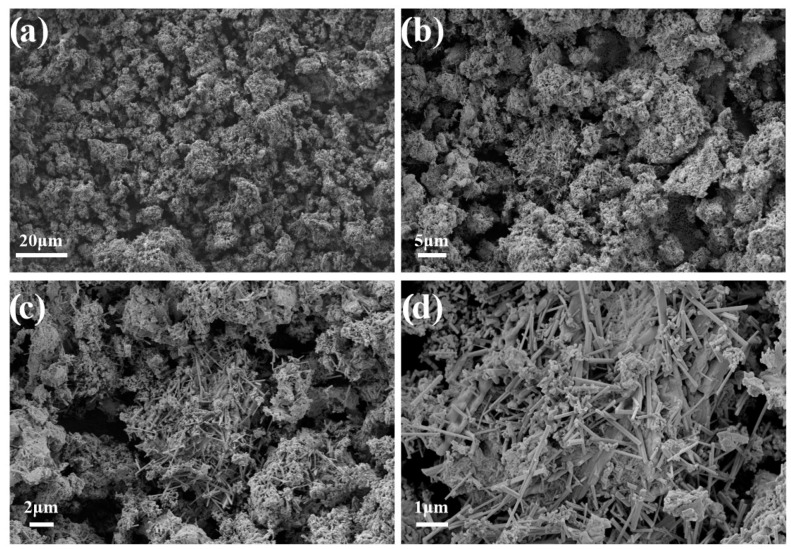
SEM images of the initial CaSiO_3_-HAp biocomposite powder.

**Figure 4 jfb-14-00259-f004:**
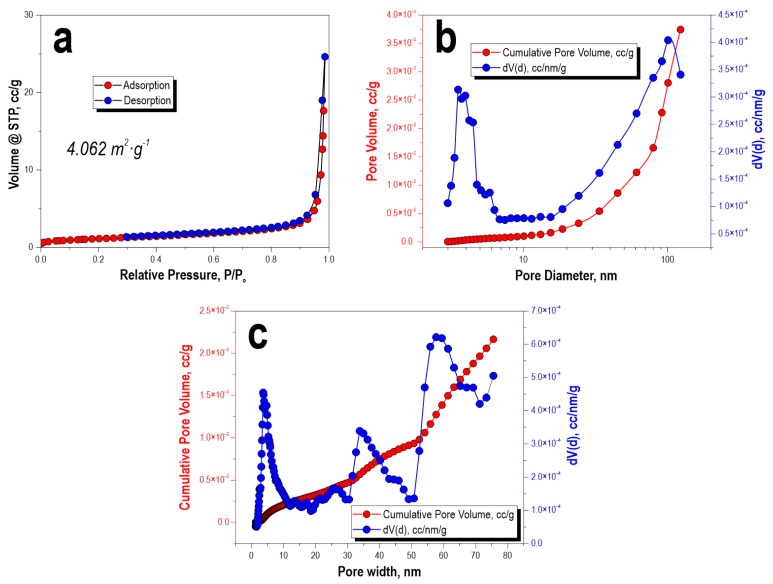
Low-temperature adsorption–desorption isotherms of nitrogen (**a**), pore-size distribution: (**b**) BJH calculation and (**c**) DFT model calculation for the initial CaSiO_3_-HAp biocomposite powder.

**Figure 5 jfb-14-00259-f005:**
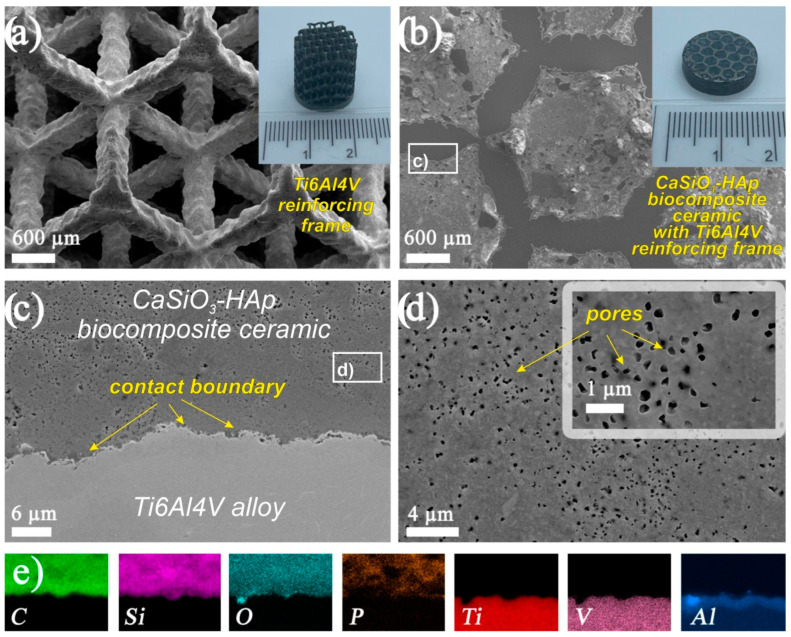
SEM images of the samples’ surfaces: (**a**) reinforcing matrix Ti_6_Al_4_V alloy; (**b**) sample CaSiO_3_-HAp biocomposite reinforced by matrix Ti6Al4V alloy; (**c**) the contact site of CaSiO_3_-HAp ceramic and Ti_6_Al_4_V alloy; (**d**) surface of CaSiO_3_-HAp ceramic; (**e**) EDS analysis of biocomposite sample surface in the contact area of ceramic and alloy. The insets show a general view of the initial titanium alloy matrix samples and the reinforced CaSiO_3_-HAp biocomposite sample.

**Figure 6 jfb-14-00259-f006:**
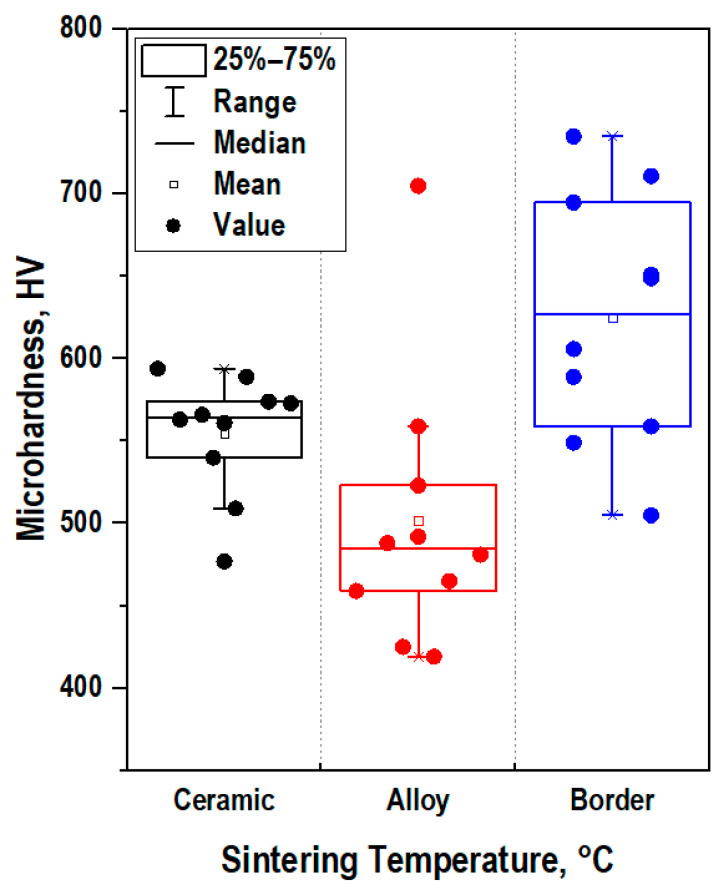
The box-and-whiskers plot of the Vickers microhardness values obtained from the surface of the CaSiO_3_-HAp biocomposite sample reinforced with Ti6Al4V alloy matrix.

**Figure 7 jfb-14-00259-f007:**
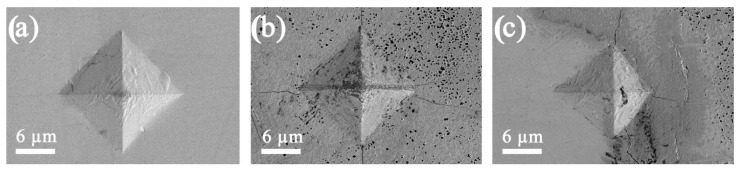
SEM images of indentor imprints on the surface of the CaSiO_3_-HAp biocomposite sample reinforced with the matrix Ti6Al4V alloy: (**a**) on the surface of the CaSiO_3_-HAp bioceramics; (**b**) on the surface of the Ti6Al4V alloy; (**c**) at the contact surface of the alloy and bioceramics.

## Data Availability

Not applicable.

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
