# Peer review of "CaSiO_3_-HAp Metal-Reinforced Biocomposite Ceramics for Bone Tissue Engineering"

_jfb, 2023, doi:10.3390/jfb14050259_

Round 1

Reviewer 1 Report

The article entitled: “CaSiO3-HAp metal-reinforced biocomposite ceramics for bone tissue engineering” is a very interesting approach to biomaterial engineering development.

In fact, just a few remarks can be made due to my personal technical limitations.

1.      I suggest replacing “etc.” to a more elegant term.

2.      Authors stated that: “This, in turn, will enhance the osseointegration properties of the ceramic metal implant.” In spite of a good theoretical support, authors can not say “will enhance”. I suggest replacing to “may enhance”.

3.      Figures are well-designed, but personally I think that yellow letters in Figure 1 were not the best choice. In Figure 6 there is a Russian word in the caption.   

Author Response

Thanks to the respectful Reviewer for his comments!

Comment #1. I suggest replacing “etc.” to a more elegant term. 

Response to comment #1: Thanks to the Reviewer for the suggestion! Corrections were made to the text.

Comment #2. Authors stated that: “This, in turn, will enhance the osseointegration properties of the ceramic metal implant.” In spite of a good theoretical support, authors can not say “will enhance”. I suggest replacing to “may enhance”. 

Response to comment #2: Thanks to the Reviewer for the suggestion! Corrections were made to the text.

Comment #3. Figures are well-designed, but personally I think that yellow letters in Figure 1 were not the best choice. In Figure 6 there is a Russian word in the caption.

Response to comment #3: Thanks to the Reviewer for the comment! Corrections were made to the Figure 1. The color and type style are changed. The Russian word has been removed from the caption of Figure 6.

Reviewer 2 Report

I found many errors of typing as 10-4 for example.In references number 24 is a repetition of number 23.All the references should then be controlled.

Author Response

Thanks to the respectful Reviewer for the comments!

I found many errors of typing as 10-4 for example.In references number 24 is a repetition of number 23.All the references should then be controlled.

Response to comment: Thanks to the Reviewer for the comment! The text of the manuscript has been checked and revised. Duplicate references in the list have been deleted.

Reviewer 3 Report

Overall the manuscript is well-written and can be considered for publication after some improvements. Some comments to improve the manuscript are as follows.

- Please identify the corresponding author/s by star sign.

- The study is quite comprehensive and has achieved very good results. Authors should write the abstract section in more detail.

- Keywords are used to assist search engines in finding the article. Please add 3-5 related keywords.

- The authors did not explain the novelty and significance of their work in the introduction section. Moreover, this section is not cohesive. Indeed, this section is intended to "convey the core findings of the paper," i.e., reflect the best novelty of this paper in a concise form. The authors shall show the work's best novelty, such as how your research advances the state-of-the-art of the topic/area and/or how much better is your work compared with peer researchers on the same or similar topics. At the end of this section, the main objective of this study must be mentioned.

- In the introduction part, the sentence "Most traditional sintering methods do not allow for achieving these characteristics, especially in their optimal ratio in the product." needs the following reference: https://doi.org/10.1016/j.matchemphys.2019.01.007

- How the authors recognized the different phases in the analysis of XRD in Fig. 2? It is suggested to mention whether the phases match their respective JCPDS cards or any other method. Please also revise the hydroxylapatite term inside the figure. It should be hydroxyapatite.

- Did the authors carry out wettability tests? As is well known, among the various surface properties, hydrophilicity is closely related to cell adhesion, as cell proliferation and differentiation have been shown to increase on the surface of highly hydrophilic materials.

- EDS measurements on light elements such as C, and O are only qualitative, not quantitative. Please make this clear. Also, indicate if the measurement (Fig. 5) was performed in the same area, or?

- Acronyms can be introduced into the text and must be defined at first use. Please check all of the manuscripts.

- There are some errors in the figure captions, for example, in Fig 6. Please revise them.

-There are some formatting mistakes in the references section, I suggest the authors check and correct them. For example, there are incomplete references or erroneous data, others with typos in the journal name, or chemical formulas in the title, for example, ref. 33.

Author Response

Thanks to the respectful Reviewer for the comments!

Overall the manuscript is well-written and can be considered for publication after some improvements. Some comments to improve the manuscript are as follows. 

 - Please identify the corresponding author/s by star sign.

Response to comment: Thanks to the Reviewer for the comment! The corresponding author is noted.

 - The study is quite comprehensive and has achieved very good results. Authors should write the abstract section in more detail.

Response to comment: Thanks to the Reviewer for the comment! Abstract of the manuscript expanded.

- Keywords are used to assist search engines in finding the article. Please add 3-5 related keywords. 

Response to comment: Thanks to the Reviewer for the comment! Keywords have been added to the manuscript. 

- The authors did not explain the novelty and significance of their work in the introduction section. Moreover, this section is not cohesive. Indeed, this section is intended to "convey the core findings of the paper," i.e., reflect the best novelty of this paper in a concise form. The authors shall show the work's best novelty, such as how your research advances the state-of-the-art of the topic/area and/or how much better is your work compared with peer researchers on the same or similar topics. At the end of this section, the main objective of this study must be mentioned.

Response to comment: Thanks to the Reviewer for the comment! The introduction presents the current state of scientific research on using wollastonite-based biocomposites for bone regeneration. A total of 37 references from the literature are cited in the introduction, more than half of them from the last five years. The scientific novelty of this work is the use of a new original composition in a mixture of wollastonite and HAp for the reinforcement of its structure. In addition, the interface between the ceramic and the alloy has been studied, which has not been reported in the known literature. The last paragraph of the introduction concludes with the statement of the objective of the research work. 

- In the introduction part, the sentence "Most traditional sintering methods do not allow for achieving these characteristics, especially in their optimal ratio in the product." needs the following reference: https://doi.org/10.1016/j.matchemphys.2019.01.007.

Response to comment: Thanks to the Reviewer for the comment! Thanks to the esteemed Reviewer for the valuable reference to a very valuable article! The reference is given in the text.

- How the authors recognized the different phases in the analysis of XRD in Fig. 2? It is suggested to mention whether the phases match their respective JCPDS cards or any other method. Please also revise the hydroxylapatite term inside the figure. It should be hydroxyapatite.

Response to comment: Thanks to the Reviewer for the comment! Figure 2 has been revised. The legend of the updated figure shows the phases of the composition with the card numbers from the PDF-2 XRD database. Since the database does not contain the Ti6Al4V alloy, the main peaks correspond to the titanium peak due to the titanium being the main peak in the alloy. A typo in the legend has been corrected. 

- Did the authors carry out wettability tests? As is well known, among the various surface properties, hydrophilicity is closely related to cell adhesion, as cell proliferation and differentiation have been shown to increase on the surface of highly hydrophilic materials.

Response to comment: Thanks to the Reviewer for the comment! An absolutely fair comment. The paper is concerned with the materials science part of the project. However, the experiment mentioned by the Reviewer is scheduled. We are currently carrying out the in vitro studies (cytology, morphometry and SBF) and the in vivo studies. The material obtained is now already embedded in the soft tissues of the lamb (the results are expected in 6 months). This new part will be dedicated to the medical block, which is the responsibility of our colleagues from the medical corps in our research team.

 - EDS measurements on light elements such as C, and O are only qualitative, not quantitative. Please make this clear. Also, indicate if the measurement (Fig. 5) was performed in the same area, or?

Response to comment: Thanks to the Reviewer for the comment! We agree with the distinguished Reviewer that the EDS method is not sufficiently accurate for the measurement of the concentrations of carbon and oxygen. This paper does not present quantitative results for carbon and oxygen. Carbon and oxygen are only shown on the EDS elemental distribution maps (Figure 5). These maps characterise the presence of carbon and oxygen only qualitatively.

 - Acronyms can be introduced into the text and must be defined at first use. Please check all of the manuscripts.

Response to comment: Thanks to the Reviewer for the comment! Abbreviations have been introduced for appropriate cases.

 - There are some errors in the figure captions, for example, in Fig 6. Please revise them. - 

Response to comment: Thanks to the Reviewer for the comment! The caption of figure 6 has been corrected.

 -There are some formatting mistakes in the references section, I suggest the authors check and correct them. For example, there are incomplete references or erroneous data, others with typos in the journal name, or chemical formulas in the title, for example, ref. 33.

Response to comment: Thanks to the Reviewer for the comment! The list of references has been corrected. The references in the text are prepared using the Mendeley referencing manager according to the JFB citation style, which complies with MDPI reference list formatting requirements. All sources have a complete set of references, including DOI and ISBN.

Reviewer 4 Report

The article ‘CaSiO3-HAp metal-reinforced biocomposite ceramics for bone

tissue engineering’ is an interesting manuscript. The work describes the possibility of using wollastonite (CaSiO3) reinforced with synthetic hydroxyapatite as a bioceramic base in the considered ceramic-metal composite, so this article fits the subject of the Journal of Functional Biomaterials. However the manuscript include none references in ‘Result and discussion section'. The disscussion in a scientific paper should be based on references. In the view of above paper should be rejected

Additionaly some minor flaws are listed below:

1.     Three to ten pertinent keywords need to be added after the abstract.

2.     There is lack of information about reagents grade and suppliers.

3.     The abbreviations of method used in a section ‘Characterization methods’ should be explained when used first time in the text f.e. scanning electron microscopy (SEM).

4.     In a section ‘Characterization methods’ XRD method is described but in the abstract the authors say that the phase and elemental compositions, structure, and morphology of the initial were studied by X-ray fluorescence. XRD is X-ray diffraction analysis. XRF is XRF is X-ray fluorescence.

5.     If analyzed samples was sputtered before making SEM images? If yes informations about it should be added. There is lack information about magnification.

6.     The crack initiation load is abbreviated as FQ in equation 1 but in the text  below equation as Fq.

7.     Figure 4a - should be ‘adsorption’ and ‘desorption’.

8.     Sentence ‘According to XRF data, the composition of the initial sintered biocomposite powder did not change under SPS conditions in the region of the studied temperature and was represented by crystalline phases of wollastonite and HAp (Fig. 2, curve 2)’ is releated to Fig. 2 which presents X-ray diffraction patterns not XRF method results.

Author Response

The article ‘CaSiO3-HAp metal-reinforced biocomposite ceramics for bone tissue engineering’ is an interesting manuscript. The work describes the possibility of using wollastonite (CaSiO3) reinforced with synthetic hydroxyapatite as a bioceramic base in the considered ceramic-metal composite, so this article fits the subject of the Journal of Functional Biomaterials. However the manuscript include none references in ‘Result and discussion section'. The disscussion in a scientific paper should be based on references. In the view of above paper should be rejected.

Additionaly some minor flaws are listed below:

Comment #1. Three to ten pertinent keywords need to be added after the abstract.

Response to comment #1: Thanks to the Reviewer for the comment! Keywords have been added to the manuscript.  

Comment #2. There is lack of information about reagents grade and suppliers.

Response to comment #2: Thanks to the Reviewer for the comment! Reagents information is added. 

Comment #3. The abbreviations of method used in a section ‘Characterization methods’ should be explained when used first time in the text f.e. scanning electron microscopy (SEM). 

Response to comment #3: Thanks to the Reviewer for the comment! Abbreviations have been introduced for appropriate cases.

Comment #4. In a section ‘Characterization methods’ XRD method is described but in the abstract the authors say that the phase and elemental compositions, structure, and morphology of the initial were studied by X-ray fluorescence. XRD is X-ray diffraction analysis. XRF is XRF is X-ray fluorescence. 

Response to comment #4: Thanks to the Reviewer for the comment! There was a typographical error in the method abbreviation during the translation of the text into English. Corrected. 

Comment #5. If analyzed samples was sputtered before making SEM images? If yes informations about it should be added. There is lack information about magnification. 

Response to comment #5: Thanks to the Reviewer for the comment! In order to compensate for surface charging, a gold layer of up to 10 nm was sputtered on the samples. The magnification information is presented on the SEM images as scale bars.  

Comment #6. The crack initiation load is abbreviated as FQ in equation 1 but in the text  below equation as Fq.

Response to comment #6: Thanks to the Reviewer for the comment! The abbreviated FQ in equation 1 are corrected. 

Comment #7. Figure 4a - should be ‘adsorption’ and ‘desorption’.

Response to comment #7: Thanks to the Reviewer for the comment! The typo in the figure 4 legend has been corrected.

Comment #8. Sentence ‘According to XRF data, the composition of the initial sintered biocomposite powder did not change under SPS conditions in the region of the studied temperature and was represented by crystalline phases of wollastonite and HAp (Fig. 2, curve 2)’ is releated to Fig. 2 which presents X-ray diffraction patterns not XRF method results.

Response to comment #8: Thanks to the Reviewer for the comment! There was a typographical error in the method abbreviation during the translation of the text into English. Corrected.

Reviewer 5 Report

The study has scientific merit and has many material characterization techniques.  The introduction has many relevant citations however the results and discussion section does not relate any of your findings to these studies.  Please compare and contrast your results to these studies.  I would also like to see the porosity characterized with pore diameter, total pores, and percent porosity (total pores/total area).

Author Response

The study has scientific merit and has many material characterization techniques.  The introduction has many relevant citations however the results and discussion section does not relate any of your findings to these studies.  Please compare and contrast your results to these studies.  I would also like to see the porosity characterized with pore diameter, total pores, and percent porosity (total pores/total area).

Response to comment: Thanks to the Reviewer for the comment!

The format of this paper is a short communication. The introduction presents the very current state of scientific research on using wollastonite-based biocomposites for bone regeneration (a total of 37 references from the literature are cited in the introduction, more than half of them from the last five years). The scientific novelty of this work is the use of a new original composition in a mixture of wollastonite and HAp for the reinforcement of its structure. In addition, the interface between the ceramic and the alloy has been studied, which has not been reported in the known literature. The last paragraph of the introduction concludes with the statement of the objective of the research work. 

Regarding the comment on the porosity question. These data were obtained earlier and have been published in a series of papers [10.1016/j.ceramint.2021.04.258 , 10.3390/jfb11020041, 10.1016/j.powtec.2020.04.040]. The powder presented in this paper should resorb in the body, initiate osteosynthesis and the role of matrix for directed bone growth is a metal matrix based on Ti6Al4V alloy.  

Round 2

Reviewer 4 Report

In the previous review, I wrote:

"However the manuscript includes no references in the ‘Result and Discussion section. The discussion in a scientific paper should be based on references. In the view of the above paper should be rejected"
The corrected version of the manuscript still does not include any references in the experimental part. In my opinion, the paper should be rejected.

Author Response

Thank you for the opportunity to make a correction. We have expanded the results discussion block. And we've added new links. The new block is highlighted.